# Blood, Hair and Feces as an Indicator of Environmental Exposure of Sheep, Cow and Buffalo to Cobalt: A Health Risk Perspectives

Muhammad Iftikhar Hussain [1,2,*] , Zafar Iqbal Khan [3], Majida Naeem [3], Kafeel Ahmad [3], Muhammad Umer Farooq Awan [4], Mona S. Alwahibi [5] and Mohamed Soliman Elshikh [5]

1 Department of Plant Biology & Soil Science, Campus Lagoas Marcosende, Universidade de Vigo, 36310 Vigo, Spain
2 CITACA, Agri-Food Research and Transfer Cluster, Campus da Auga, Universidade de Vigo, 32004 Ourense, Spain
3 Department of Botany, University of Sargodha, Sargodha 40100, Pakistan; zafar.khan@uos.edu.pk (Z.I.K.); arwa.fatima@gmail.com (M.N.); kafeeluaf@yahoo.com (K.A.)
4 Department of Botany, Government College University, Lahore 54000, Pakistan; bibi04572@gmail.com
5 Department of Botany and Microbiology, College of Science, King Saud University, Riyadh 11451, Saudi Arabia; malwhibi@ksu.edu.sa (M.S.A.); melshikh@ksu.edu.sa (M.S.E.)
* Correspondence: mih786@gmail.com

**Abstract:** Exposure to toxic metals (TMs) such as cobalt (Co) can cause lifelong carcinogenic disorders and mutagenic outcomes. TMs enter ground water and rivers from human activity, anthropogenic contamination, and the ecological environment. The present study was conducted to evaluate the influence of sewage water irrigation on cobalt (Co) toxicity and bioaccumulation in a soil-plant environment and to assess the health risk of grazing livestock via forage consumption. Cobalt is a very necessary element for the growth of plants and animals; however, higher concentrations have toxic impacts. Measurement of Co in plant, soil and water samples was conducted via wet digestion method using an atomic absorption spectrophotometer. The Co pollution severity was examined in soil, forage crops (*Sorghum bicolor* Kuntze, *Sesbania bispinosa* (Jacq.) W. Wight, *Cynodon dactylon* (L.) Pers., *Suaeda fruticosa* (L.) Forssk. and *Tribulus terrestris* L.) in blood, hair and feces of sheep, cow and buffalo from district Toba-Tek-Singh, Punjab, Pakistan. Three sites were selected for investigation of Co level in soil and forage samples. Highest concentration of Co was 0.65 and 0.35 mg/kg occurring in *S. bicolor* at site I. The sheep blood, cow hair and sheep feces samples showed highest concentrations of 0.545, 0.549 and 0.548 mg/kg, respectively at site I and site II. Bioconcentration factor, pollution load index, enrichment factor and daily intake were found to be higher (0.667, 0.124, 0.12 and 0.0007 mg/kg) in soil, *S. bicolor*, *S. fruticosa* and in buffalo, respectively, at site I. It was concluded that forage species irrigated with wastewater are safe for consumption of livestock. However, though the general values were lower than the permissible maximum limit, it was observed that the bioaccumulation in the forage species was higher. Therefore, soil and food chain components should be avoided from trace metal contamination, and other means of nonconventional water resources should be employed for forages irrigation.

**Keywords:** heavy metals; metal toxicity; plant-soil environment; effluents; livestock

## 1. Introduction

The agriculture sector needs a significant amount of fresh water to increase agriculture production (vegetables, fruits, cereals, oil seed crops, legumes and forages) in order to achieve a country's food security and resilience [1]. Due to shortage of water in the arid and semi-arid regions like Pakistan, there is an urgent need to look for alternate water resources (treated waste water, rain harvesting, small ponds construction in drought prone areas [1,2]. Therefore, alternate water resources can avoid a threat to freshwater water

resources and also helps to protect the groundwater resources. However, several factors might influence farmers' attitudes towards the safe reuse of agriculture and drainage water for cultivation of forage crops. A detailed study about different regions, farmers attitude, and livestock feed irrigation that indicates different responses from different regions has already been published [3,4].

Environmental pollution is the major outcome of the urbanization, industrialization, water scarcity, drought and unchecked disposal sites that can lead to biodiversity loss and a decrease in quality of water, soil and atmosphere. An increase in the biodiversity loss as a result of use of untreated wastewater and unchecked sewerage sludge dumping will enhance bioaccumulation of contaminants and heavy metals (HMs) in soil, plants and animals [5,6]. The substantial increase of hazardous metals via food consumption will result in a bad impact on human health [7]. Inorganic contaminants include heavy metals like Al, Zn, Cd, Se, Pb, Hg, As, Cr, Bi and are major causes of concern because metals cannot be degraded but can be changed from one form to other through oxidation reduction reactions [8]. These are introduced into soil through mining, nuclear processing and industrial manufacturing. Heavy metals are a vital source of environment degradation and biodiversity loss, to which plants, and in particular vegetation, seem to reply specifically [9,10]. It will demolish the quality of human life with significant increase in skin, bone and kidney failure disorders, and thus it adversely affects human health when ingested as food for present and future generations [2]. Heavy metals attach with proteins which are not made for them by dislodging novel metals from their basic confining regions causing separating of cells and, in the end, toxicity. Past research has found that oxidative damage to characteristic macromolecules is essentially a direct result of a limiting of profound metals to the DNA and nuclear proteins [11]. The continuous use of wastewater will deposit a significant amount of heavy metals that will ultimately become part of plant-soil ecosystems, and thus pose serious toxic effects to both the environment and human health. Many studies documented that several human sicknesses are directly correlated with metal intoxication that enters in the food chain through the water–soil–plant ecosystem [11,12]. Reports published by several international agencies like US Environmental Protection Agency (EPA) documented that several trace elements like, Pb, As, Cd, Hg, Cr ranked among the priority toxic elements for public health and has been classified as human carcinogens [13].

Cobalt (Co) is not an essential micronutrient for plants and animals. However, its essence in higher plants has not been seen now. Water pollution and extensive fertilizer application practices poses a serious risk to human health and the terrestrial's ecosystems, via metals, pesticides, and toxic substances into the environment, a problem often underestimated by policymakers and farmers alike [1,2]. Cobalt is a vital trace metal, required by some specific plants for certain biochemical attributes and metabolism [14]. Some reports showed its importance in the legumes for root functions and biological nitrogen fixation via root nodules [15]. Small quantities of Co might be beneficial for plants and animals; however, higher concentration is toxic for the plant–soil–water–animal nexus that can produce adverse effects. The Co toxicity depends on soil biology, physiological properties of certain plant species and crop types [16]. Accumulation of Co in plant tissues causes reduction in plant growth, dry biomass, water and nutrient uptake, degradation of protein contents, and photosynthetic pigments [17,18]. Several literature reports showed that some plants accumulate a significant portion of Co from soils irrigated with wastewater, and hence, there is a high dietary intake of this element through food crop consumption [19]. HMs are absorbed from wastewater and passed through animals to humans, and thus, the crop plants and forages might be considered as corridors for HMs entry into humans [20].

The livestock sector is a significant sponsor to Gross Domestic Product (GDP) of several arid and semi-arid countries of the world, including Pakistan [11,12,21]. In fact, >33% of agriculture lands is utilized by growers for cultivation of forages and grasses of high nutritional value [22]. In several Asian and African countries, more than 30% of the available land has been dedicated for cultivation of green fodder crops which are

an extensive example of a fresh water consumer [22,23]. According to several literature reports, approximately 85 L of water has been consumed to produce 1 kg of forage crops, which is expensively higher than to produce vegetables that only require as little as 43 L of water per kg [24]. Meanwhile, a large quantity of fresh water has been used for production of green fodders and perennial forage grasses ultimately consumed by livestock, and this quantity of water is around 29% of water used in agriculture [9,12].

Domesticated animals, chickens and birds are among the important nature creatures that can be utilized for biomonitoring of ecological contamination due to trace metals in the plant–soil–water environment [5,6]. Biomonitoring of the birds and ruminants offers lifetime expectancy, tremendous accessibility of essential information and moderately basic inspecting methodology [3]. Meanwhile, cattle slurries have lethal impact due to the presence of these metals that are unsafe to human wellbeing and remain in soil for a long time [25]. Trace elements present in cows' milk were reported by Ward and Savage [26] because these cows were fed with grasses and forage crops irrigated with wastewater. In Pakistan, several authors reported the presence of heavy metals (Cd, Cr, Ni, Pb and As) in cows' and goats' milk. These studies showed that the presence of HMs in the animals' milk was mainly attributed to the HMs presence and translocation from forage crops cultivated on contaminated soil [27,28]. Meanwhile, other authors [29,30] showed that consumption of milk from cows fed with contaminated forages (via wastewater irrigation) had high levels of Pb, Zn, Cr and Ni in their blood. Therefore, the present research was conducted with the aim to determine the heavy metal, Co, in the body tissues of cattle that were fed through forage crops that were grown on soil irrigated with wastewater. The current study aimed to quantify the heavy metals (cobalt) concentrations in the blood, hair and feces of buffalo, cow and sheep from Toba-Tek-Singh (Punjab, Pakistan) to assess feasibility as a biomonitoring agent for the environmental hazards assessment of heavy metals pollution via forage crops (*Sorghum bicolor* Kuntze, *Sesbania bispinosa* (Jacq.) W. Wight, *Cynodon dactylon* (L.) Pers., *Suaeda fruticosa* (L.) Forssk. and *Tribulus terrestris* L.

## 2. Materials and Methods

### 2.1. Study Area

Toba-Tek-Singh Region is located in Southern Punjab (Pakistan) and geographically arranged in Rechna Doab between Chenab Stream and Ravi Waterway. Toba-Tek-Singh exhibit 30°33′ to 31°2′ N and the longitude 72°08′ to 72°48′ E. Dry climate extends from April to October, while May, June and July are the most sizzling months. The mid-year season temperature is in the range of 42 °C and 29 °C, while cooler months include December, January and February. Annual precipitation is mostly around 158 mm. The soil of Toba-Tek-Singh Region is mostly alluvial plain, and many crops—wheat, maize, sunflower, rice, cotton and several kinds of forage crops—are grown. Chenab and Ravi supply most of the water through small streams and link canals via water channels up to the farmer's field. The local water resources are not adequate to restore the ground water for drinking and water framework reason, so the streams are the essential source to invigorate the groundwater assets [31]. Study was led in Toba-Tek-Singh.

### 2.2. Sample Collection

Testing of soil, blood and hair was completed in the summer season. From each destination, summer inspection used to be executed in August 2019. For the present study, three different sites were selected at Toba-Tek-Singh (Figure 1). The study sites were selected at random and 1 km away from roads. Samples, along with 3 replicates, were collected as per procedure [32]. Soil and plant samples were taken sealed in label paper bags. The soil samples were chosen from 3 different site of soil of *Sorghum bicolor*, soil of *Sesbania bispinosa*, soil of *Cynodon dactylon,* soil of *Suaeda fruticosa* and soil of *Tribulus terrestris*. The forage chosen for this experiment includes *Sorghum bicolor*, *Sesbania bispinosa*, *Cynodon dactylon*, *Suaeda fruticosa* and *Tribulus terrestris*. Meanwhile, the blood, hair and feces samples of cow, buffalo and sheep were collected.

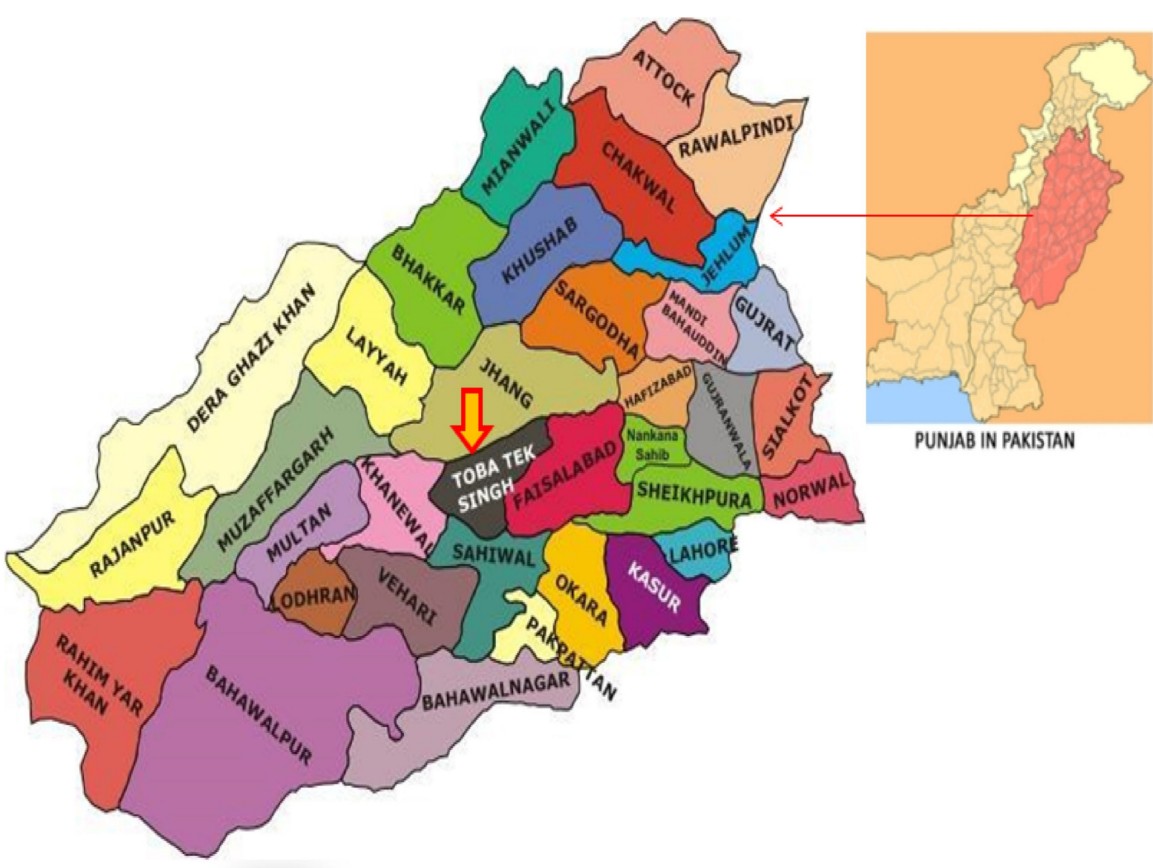

**Figure 1.** Study map of Toba-Tek -Singh District, Punjab Pakistan.

The soil samples were collected simultaneously at 5, 10, 15 and 20 m away from highway at a soil depth of 20 cm. [12]. Forage and soil samples were dried in stove at 72 °C for 48 h to achieve a steady dry weight. Sampling was done in August 2019. Soil samples were taken from different depths of field soils having forages. The same forage plots were used as grazing units for selected animals. The oven-dried soil samples were subjected to digestion and analysis. The healthy animals were selected for blood plasma sampling and to determine the heavy metal (Co). Specimen of blood (15 mL) was collected from the jugular vein of selected animals with a help of heparin needles to avoid blood coagulation [12]. Plasma was centrifuged at 2500 rpm for 2 min to make plasma parcel. The serum was taken to the laboratory in a chilled box and maintained frozen at −20 °C before investigation [32].

### 2.3. Soil Sample Digestion

The soil samples were dried in stove for 72 h followed by grinded into fine powder. The powdered soil sample of 2 g was added to the digestion tube. Add 20 mL concentrated $H_2SO_4$ into the tube. Run the digestion chamber for 30 min. Further add 10 mL $H_2O_2$ to assist digestion. And again run the chamber until colorless solution was obtained. Solution was filtered and makes a final volume 60 mL through distilled water addition. The final volume of sample was stored in glass tubes for further metal analysis [12].

### 2.4. Forage Sample Digestion

Forage sample were dried in an oven for 72 h to eliminate moisture content and afterward crushed with the help of a blender. The 1 g of forage samples were digested with $H_2SO_4$ and $H_2O_2$ (4:2) at a temperature of 250 °C for 3–4 h till the solution became colorless and thick white fume arose in the flask. The mixture was incubated, cleaned with



distilled water, passed through filter paper and diluted the sample to make the volume up to 50 mL [12].

### 2.5. Blood Sample Digestion

The anticoagulant blood was centrifuged. At that point 2 mL of blood plasma were mixed with 2 mL of $H_2SO_4$ and combined sample were left in the lab for entire night for digestion. All the natural tissues were broken down and solution of sample was digested by warming at 120 °C. At this stage, 2 mL of $H_2O_2$ has been added to improve the digestion by degradation. The treated samples were cooled and the processed digested samples were diluted up to 50 mL using distilled water and set in glass tubes for assessment. Solid metal concentrations were determined utilizing the Atomic Absorption Spectrophotometer (AAS) (Shimadzu double beam AA-6300 and Perkin Elmer Analyst 400) [12].

### 2.6. Atomic Absorption Spectrum

Heavy metal cobalt (Co) concentration was analyzed by atomic absorption spectrum (Shimadzu double beam AA-6300 and Perkin Elmer Analyst 400). The quality assurance was achieved by measuring natural matrix certified reference material (CRM-1570) and measuring duplicates for each batch of samples to ensure the stability of the outcomes. The samples were handled cautiously to inhibit pollution. Double distilled water was used during the evaluation and glass was methodically washed. An analysis was carried out to validate different steps by homogenizing the examined samples with a varying quantity of standard solutions.

### 2.7. Statistical Analysis

Data was analyzed through analysis of variance (ANOVA), and difference between treatments means were computed at $p < 0.05$ according to post hoc Tukey's HSD (honestly significant difference) test. General linear model was used for data analysis using software (SPSS (version 21.00) for Windows (SPSS Inc., Chicago, IL, USA), and graphs were prepared using means and S.E. of the respective traits using Microsoft Excel package.

### 2.8. Pollution Load Index (PLI)

PLI was estimated by equation given by Liu et al. [33].
PLI = 'metal content in examined soil sample/metal concentration in reference soil'.
Reference value of Pb and Co in soil is 8.15 and 9.1 mg/kg [33].

### 2.9. Bioconcentration Factor (BCF)

It was estimated by following formula as reported by Cui et al. [34].
BCF = [M] Fodder samples/[M] Soil sample.

### 2.10. Daily Intake of Metal (DIM)

Metals enter in body of organisms through diverse pathways, through skin contact, during breathing or by consuming contaminated fodder [35].
DIM = C metal × F conversion factor × D food intake/B average weight of sheep DIM by ingesting of forages is 1.51 (kg per sheep-1) and normal weight of sheep is 45 kg was used as reported by Bonnet et al. [36].
Conversion factor of 0.085 was applied to change green plant mass to dry weight [37].
Tolerable daily intake limit of Pb and Cu is 0.21 and 3.01 (mg kg$^{-1}$day$^{-1}$).

### 2.11. Health Risk Index (HRI)

Oral reference dose value for Pb and Cu is 0.0035 and 0.04 (mg kg$^{-1}$day$^{-1}$) reported by World Health Organization [38].
Health risk index = Daily intake of Metal/RfD [13].

### 2.12. Enrichment Factor (EF)

It was determined by formula of Buat-Menard and Chesselet [39].

EF = [M] Fodder E/[M] Soil E/[M] Fodder S/[M] Soil S Average absorption of Pb and Cu in forages is 2 and 10 mg/kg and in soil is 8.15 and 8.39 mg/kg, respectively [40].

## 3. Results

### 3.1. Ecological Risk Assessment through Soil Analysis

Highest concentration of Co was found in soil of *S. bicolor* at site I, while lowest concentration was observed in soil of *S. fruticosa* at site I. Table 1 demonstrates the analysis of variance, degree of freedom and treatment details and their interaction. The concentration of Co in soil of all forages ranged from 0.42 mg/kg to 0.65 mg/kg (Figure 2). In soil sample, highest mean concentration of Co was 0.65 mg/kg occurring in *S. bicolor* at site I and lowest mean concentration was 0.42 mg/kg occurred in *S. fruticosa* at site I. Concentration of Co in soil sample ranged from 0.42 mg/kg to 0.65 mg/kg. The order of Co concentration at site in soil was *S. bicolor* > *C. dactylon* > *S. bispinosa* > *T. terrestris* > *S. fruticosa* (Figure 2). At site II, the Co concentration order in soil was *C. dactylon* > *S. bispinosa* > *T. terrestris* > *S. bicolor* > *S. fruticosa*. At site II, Co concentration order in soil was *C. dactylon* > *T. terrestris* > *S. bicolor* > *S. fruticosa* > *S. bispinosa*. Analysis of variance showed the non-significant effect of Co at site, forage and site × forage (Figure 2).

**Table 1.** Variance analysis of Co metal in soil.

| Source | Degree of Freedom | Mean Square |
|---|---|---|
| Site | 2 | 0.009 [ns] |
| Soil | 4 | 0.02 [ns] |
| Site × soil | 8 | 0.014 [ns] |

[ns] insignificant.

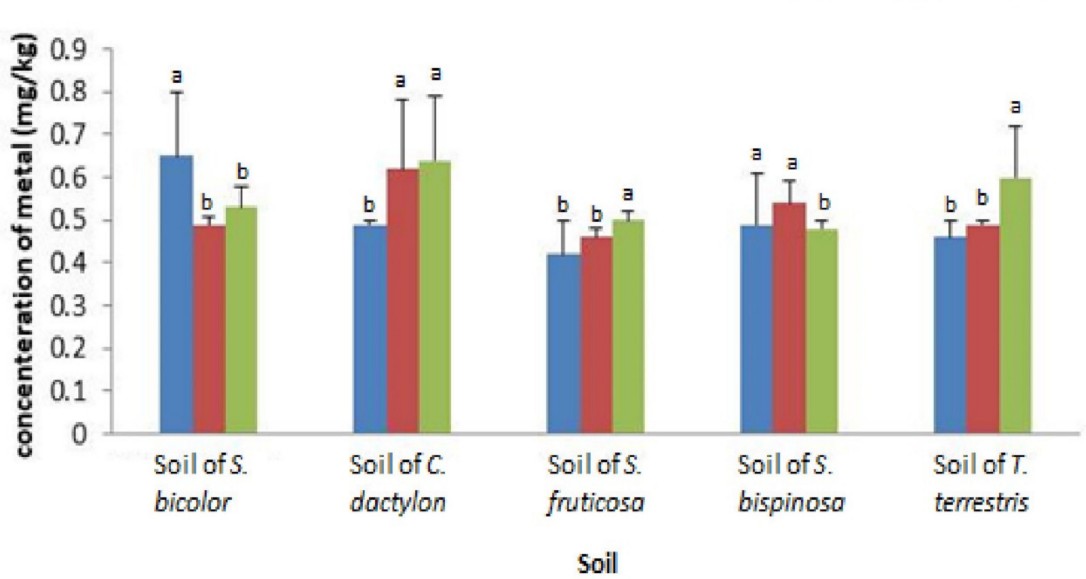

**Figure 2.** The concentration of heavy metal Co in soil. Every bar represents the mean (±S.E.) of three replicates. Means followed by different letters are significantly different ($p < 0.05$) according to Tukey's HSD test.

### 3.2. Contamination of Co Toxicity on Forage Growth

In forage sample, highest mean concentration of Co was 0.35 mg/kg occurring in *S. bicolor* at site I and lowest mean concentration was 0.16 mg/kg occurring in *C. dactylon* at site I (Figure 3). Table 2 demonstrates the analysis of variance, degree of freedom and treatment details and their interaction. Concentration of Co in forage sample ranged from 0.16 mg/kg to 0.35 mg/kg. The order of Co concentration at site in forage was *S. bicolor* > *S. fruticosa C. dactylon* > *S. bispinosa* > *T. terrestris*. At site II, Co concentration order in forage was *S. bispinosa* > *T. terrestris* > *S. bicolor* > *S. bispinosa* > *S. fruticosa* > *C. dactylon* (Figure 3). At site II, Co concentration order in soil was *C. dactylon* > *S. fruticosa* > *S. bispinosa* > *T. terrestris* > *S. bicolor*. Analysis of variance showed the significant effect of Co at site and site × Animal, while non-significant effect of Co was present in Animal, Source, Site × Source, and Animal × Source and Site × Animal × Source (Table 2, Figure 3).

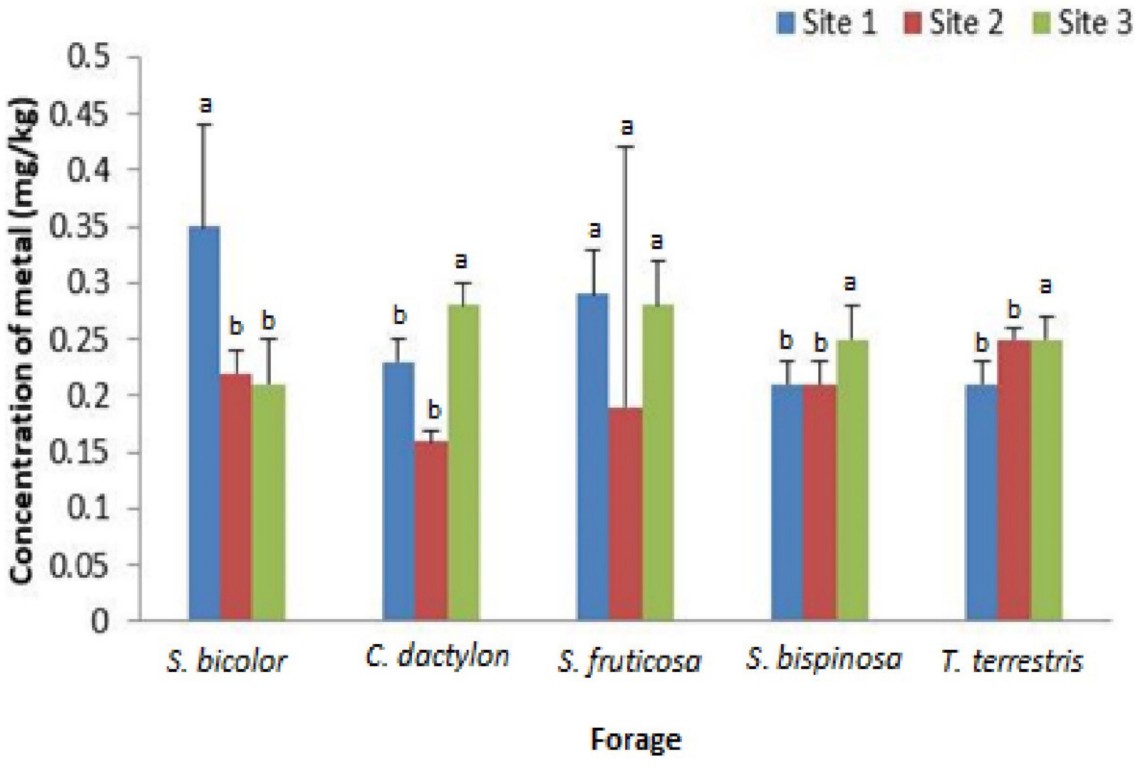

**Figure 3.** The concentration of heavy metal Co in the forages. Every bar represents the mean (±S.E.) of three replicates. Means followed by different letters are significantly different ($p < 0.05$) according to Tukey's HSD test.

**Table 2.** Analysis of Variance for Co metal in forages.

| Source | Degree of Freedom | Mean Square |
|---|---|---|
| Site | 2 | 0.013 ns |
| Forage | 4 | 0.003 ns |
| Site × forage | 8 | 0.007 ns |

ns insignificant.

### 3.3. Metal Concentration of Co in Animal Blood, Hair and Feces

Highest concentration of Co was found in sheep blood at site II, while lowest concentration was observed in Buffalo blood at site I (Tables 3 and 4). Concentration of Co in animal blood ranged from 0.404 mg/L to 5.45 mg/L. Highest concentration of Co was 0.549 mg/kg found in cow hair at site I, while lowest concentration 0.389 mg/kg was observed in sheep hair at site III. Concentration of Co in animal hair ranged from 0.389 mg/kg to 2.549 mg/kg (Table 4). Concentration of Co in animal feces ranged from 0.398 mg/kg to

0.548 mg/kg. Highest concentration of Co (0.548 mg/kg) was found in sheep feces at site I and lowest concentration (0.398 mg/kg) was observed in Buffalo feces at site II (Table 4).

**Table 3.** Analysis of Variance for Co metal in animal blood, hair and feces.

| Source | Df | Mean Square |
|---|---|---|
| Site | 2 | 0.124 *** |
| Animal | 2 | 0.023 ns |
| Source | 2 | 0.02 ns |
| Site × Animal | 4 | 0.055 *** |
| Site × Source | 4 | 0.011 ns |
| Animal × Source | 4 | 0.008 ns |
| Site × Animal × Source | 8 | 0.007 ns |

ns insignificant. *** Significant at 0.001 levels.

**Table 4.** Concentration of heavy metal Co in animal blood, hair and feces.

| Source | Animal | Site 1 | Site 2 | Site 3 |
|---|---|---|---|---|
| Blood | Cow | 0.409 ± 0.02 | 0.482 ± 0.03 | 0.437 ± 0.02 |
| | Buffalo | 0.404 ± 0.05 | 0.485 ± 0.01 | 0.408 ± 0.02 |
| | Sheep | 0.443 ± 0.05 | 0.545 ± 0.02 | 0.416 ± 0.01 |
| Hair | Cow | 0.549 ± 0.04 | 0.522 ± 0.01 | 0.435 ± 0.01 |
| | Buffalo | 0.500 ± 0.30 | 0.458 ± 0.04 | 0.449 ± 0.02 |
| | Sheep | 0.484 ± 0.05 | 0.526 ± 0.01 | 0.389 ± 0.01 |
| Feces | Cow | 0.526 ± 0.04 | 0.491 ± 0.04 | 0.420 ± 0.02 |
| | Buffalo | 0.494 ± 0.02 | 0.398 ± 0.01 | 0.418 ± 0.01 |
| | Sheep | 0.548 ± 0.04 | 0.439 ± 0.01 | 0.406 ± 0.02 |

*3.4. Human Health Risk Assessment from Dietary Uptake*

3.4.1. Bioconcentration Factor in Co Metal

Bioconcentration factor recorded for Co was found to be higher, 0.667 mg/kg, in soil of *S. fruticosa* at site I and lower, 0.258 mg/kg, in soil of *C. dactylon* at site I. Bioconcentration factor ranged from 0.258 mg/kg to 0.667 mg/kg (Table 5).

**Table 5.** Bioconcentration Factor of Co Metal in soil sample.

| BCF | Site 1 | Site 2 | Site 3 |
|---|---|---|---|
| Soil of *S. bicolor* | 0.538 | 0.449 | 0.396 |
| Soil of *C. dactylon* | 0.469 | 0.258 | 0.438 |
| Soil of *S. fruticosa* | 0.667 | 0.413 | 0.56 |
| Soil of *S. bispinosa* | 0.429 | 0.389 | 0.521 |
| Soil of *T. terrestris* | 0.457 | 0.510 | 0.417 |

3.4.2. Pollution Load Index in Co Metal

Values of pollution load index for metal Co was found higher (0.124 mg/kg) in *S. bicolor* at site I while lower (0.080 mg/kg) was observed in *S. fruticosa* at site I (Table 6).

**Table 6.** Pollution load index for Co metal.

| PLI | Site 1 | Site 2 | Site 3 |
|---|---|---|---|
| *S. bicolor* | 0.124 | 0.094 | 0.101 |
| *C. dactylon* | 0.094 | 0.119 | 0.122 |
| *S. fruticosa* | 0.080 | 0.089 | 0.096 |
| *S. bispinosa* | 0.094 | 0.103 | 0.092 |
| *T. terrestris* | 0.088 | 0.094 | 0.115 |

### 3.4.3. Enrichment Factor in Co Metal

Enrichment factor of Co was higher (0.12 mg/kg) in *S. fruticosa* at site I while lower, 0.048, in *C. dactylon* at site II. Enrichment factor was ranged from 0.048 mg/kg to 0.121 mg/kg (Table 7). DIM amount for Co *S. bicolor* was higher, 0.0007 mg/kg, at site I in buffalo and lower, 0.0002 mg/kg, at site I in sheep samples of *C. dactylon* (Table 8). The value of daily intake of metal was ranged from 0.0002 mg/kg to 0.0007 mg/kg (Table 8). Health risk index value for Co was found to be highest in *S. bicolor* at site I in buffalo and lowest in *C. dactylon* at site II in sheep (Table 9). Health risk index factor ranged from 0.0071 mg/kg to 0.0157 mg/kg (Table 9).

**Table 7.** Enrichment factor for Co.

| EF | Site 1 | Site 2 | Site 3 |
|---|---|---|---|
| *S. bicolor* | 0.098 | 0.081 | 0.072 |
| *C. dactylon* | 0.085 | 0.048 | 0.079 |
| *S. fruticosa* | 0.121 | 0.075 | 0.109 |
| *S. bispinosa* | 0.078 | 0.071 | 0.095 |
| *T. terrestris* | 0.083 | 0.093 | 0.076 |

**Table 8.** Daily intakes of metal for Co.

| DIM | Cow | | | Buffalo | | | Sheep | | |
|---|---|---|---|---|---|---|---|---|---|
| | Site 1 | Site 2 | Site 3 | Site 1 | Site 2 | Site 3 | Site 1 | Site 2 | Site 3 |
| *S. bicolor* | 0.0006 | 0.0004 | 0.0003 | 0.0007 | 0.0004 | 0.0004 | 0.0005 | 0.0003 | 0.0003 |
| *C. dactylon* | 0.0004 | 0.0003 | 0.0005 | 0.0004 | 0.0003 | 0.0005 | 0.0003 | 0.0002 | 0.0004 |
| *S. fruticosa* | 0.0005 | 0.0003 | 0.0005 | 0.0005 | 0.0004 | 0.0005 | 0.0004 | 0.0003 | 0.0004 |
| *S. bispinosa* | 0.0004 | 0.0004 | 0.0004 | 0.0004 | 0.0004 | 0.0005 | 0.0003 | 0.0003 | 0.0003 |
| *T. terrestris* | 0.0003 | 0.0004 | 0.0004 | 0.0004 | 0.0005 | 0.0005 | 0.0003 | 0.0004 | 0.0004 |

**Table 9.** Health risk index for Co.

| HRI | Cow | | | Buffalo | | | Sheep | | |
|---|---|---|---|---|---|---|---|---|---|
| | Site 1 | Site 2 | Site 3 | Site 1 | Site 2 | Site 3 | Site 1 | Site 2 | Site 3 |
| *S. bicolor* | 0.0138 | 0.0087 | 0.0083 | 0.0157 | 0.0098 | 0.0094 | 0.0119 | 0.0075 | 0.0072 |
| *C. dactylon* | 0.0091 | 0.0063 | 0.01107 | 0.0103 | 0.0072 | 0.0125 | 0.0079 | 0.0055 | 0.0096 |
| *S. fruticosa* | 0.0111 | 0.0075 | 0.01107 | 0.0126 | 0.0085 | 0.0126 | 0.0096 | 0.0065 | 0.0095 |
| *S. bispinosa* | 0.0083 | 0.0083 | 0.0099 | 0.0094 | 0.0094 | 0.0112 | 0.0072 | 0.0072 | 0.0086 |
| *T. terrestris* | 0.0082 | 0.0099 | 0.0098 | 0.0095 | 0.0112 | 0.0113 | 0.0071 | 0.0086 | 0.0085 |

## 4. Discussion

Pollution due to trace metals in the environment has received global attention. This is mainly because of easy bioaccumulation, poor degradation after entering the soil-water environment and their subsequent health risks to humans due crops and vegetables consumption [1,3,4,6,18]. Trace metals are very dangerous and are an important reason behind several human sicknesses like cancer, DNA damage, embryo and fetus problems and pregnancy complication [41,42]. According to US Environmental Protection Agency, Hg, Cd, As, Pb and Cr low level in water is highly toxic and mostly diagnosed as 0.002, 0.005, 0.01, 0.015 and 0.1 mg/L [43]. Cobalt (Co) existing in the environment can cause lung cancer [44]. Hence, using treated wastewater for agriculture activities and reducing and minimizing heavy metals contamination and soil rhizosphere bioaccumulation is of vital importance. In this regard, management strategies should include avoiding the environment contamination, via both plants, and soil route and minimize their uptake, translocation and bioaccumulation in the food chain [1,4].

Several industries like textile, paper, ghee, tanneries, sugar mills and distilleries discharge their wastes into Shah Alam River [45]. They reported trace elements concentration in the Shah Alam River 10 times greater than the World Health Organization (WHO) standard limit. Kashif et al. [46] investigated heavy metals pollution in Hudiara drain passing near Hudiara village of district Lahore, Pakistan. The index of metal pollution enhanced due to entrance of polluted waste in the drain from nebulous sources. These metals amassed in the food chain and might be precarious for human beings. The maximum permissible limit according to WHO/FAO of Co amassing in soil was indicated as 50 mg/kg [47]. The estimations of Co metal in the current examination were lower than these MPL in soil samples. According to Tahir et al. [48], the Co in soil concentration value was higher than present study.

These lower levels of metals can be because of the sandy surface of the dirt found in those locales, bringing about lower draining layers of metal and an exceptionally low SOM that could not hold substantial metals and would permit plants to ingest metal. Metals, for example Fe and Mn, are primarily of lithological cause; however, unreasonable metal substances, for example Cr, Cd, Cu, Pb, Zn [49], are the responsibility of anthropogenic activities, although the Zn is a useful microelement for humans that occurs naturally in some ancestors of cultivated wheat [50,51]. The Cr concentration in different forages was higher than the present study in Western Siberia [52]. Another study reported higher Co than the present study [53]. According to Tahir et al. [48] Co blood concentration value was higher than in the present study. The utilization of dirtied brushing feed, water and scrounge filled in defiled soil might be because of the higher Co focuses. High cobalt levels can cause lung problems in human beings, such as asthma, pneumonia and heart failure [54]. The measure of Co estimated was extensively higher than the suggested values, so it might negatively affect living organisms. Co substance in blood was lower for Li et al. than in our investigation [55]. Co contents in hair were lower according to Li et al. [55] than in our study.

Previously, in several studies [15,28,29,48,56], the trace metals such as Cd, Cr and Pb cause significant lethal effects on plant growth and contamination to plant-soil environment in some important cities of Pakistan like Peshawar, Gujranwala, Haripur, Sargodha; these toxic metals are dangerous to the ecosystem and human health via their entry into the food chain. Heavy metal contamination was evaluated in cucumber cultivated under a conventional, greenhouse and organic production system from Egypt [57]. Results showed significantly higher toxicity due to metal bioaccumulation, while Cd and Pb were present in higher concentrations than the permissible limits. Meanwhile, the conventional system was prone to more metal contamination than other production systems [57].

In a recent study, bioconcentration factor value was 0.1–1 showed for moderate accumulator plant [58]. In our finding, the value of Co in BCF was low according to Khan et al. [9]. An overall outline of contamination at each examining site is given by PLI. PLI < 1 demonstrates that there is no heavy metal defilement at the site; PLI > 1 shows,

nonetheless, that there is pollution [59]. The present findings of Co in soils of all were lower than 1, indicated that there is no metal contamination. Pollution load index of Co metal in the investigated area was high, as given by Mihali et al. [60].

Evaluation of the impact of various heavy metals on growth and yield of wheat was investigated by Singh and Aggarwal [61]. Results revealed that heavy metals inhibited wheat growth and yield but extent of inhibition varies among metals; Hg poses strong inhibition followed by Cu, Pb and Cd. Metal stress cause reduction in 1000-grain weight, spikes pot$^{-1}$ and grains spike$^{-1}$. The vegetative part had maximum accumulation of metals than the reproductive part. Heavy metals (Cd, Pb, Cu, Ni, Cr and Zn) contamination retarded wheat growth and yield. Amongst metals listed above, trace metals were most toxic, which negatively affected grain yield, activity of *Azotobacter*, protein and nitrogen contents of wheat [62]. The Co exhibit EF $\leq$ 1 in all forage samples; it was considered as background rank. EF values > 1 indicate an anthropogenic origin, while values < 1 indicate potential metal depletion or mobilization [63].

The EF of Co in the present finding, in all forage samples, was <2, which showed minimal enrichment. In our study, enrichment factor of Co value was low as compared to other studies [64,65]. According to Roggeman et al., daily intake of Co metal for cows was higher than in our study [66]. DIM of Co metal in forage consumption was low and high as given by Khan et al. [9].

There is concern about possible health consequences if the HRI reaches 1 [67–69]). In our study, health risk index value was less than 1, indicating no future health risk. Health risk index of Co metal in forage consumption was high as given by Khan et al. [9]. In another study, dust and soil samples were collected from various places along Islamabad Expressway and analyzed for metal pollution on flame atomic absorption spectrometry (FAAS). The results showed variations in accumulation of heavy metals along the Expressway, but Cd was found up to $5 \pm 1$ mg kg$^{-1}$ soil. Owing to urbanization, heavy metal pollution increased and needs to be monitored [68]. Ahmed et al. [70] analyzed 44 samples and reported $398 \pm 183$ pg 100 mL$^{-1}$ and $768 \pm 180$ pg 100 mL$^{-1}$ Cd concentration in blood of jewelers and automobile workers, respectively. According to WHO standards, Cd was present in a safe limit, but long exposure might be precarious for jewelers and automobile workers. According to reports of Bhardwaj et al. [71], the impact of Cd and Pb on physiological traits of *Phaseolus vulgaris* L. was determined. Growth parameters were reduced as metal concentrations increased, although low metal concentration had no drastic effect on percent germination while it was utterly inhibited at higher Cd concentration.

Cobalt inhibited the crop growth because it induced some physiological changes and chlorophyll pigment synthesis in plants, as reported by Akeel et al. [72]. Significant inhibition in photosynthesis, chlorophyll (*a + b*) and total protein *Phaseolus vulgaris* L. while increase in abscisic acid and proline content in leaves of Co$^{2+}$ and Zn$^{2+}$ applied plants [73]. It has also shown inhibition of root development by retarding the division of cells and preventing nutrient and water uptake and translocation [74]. The composition of sewage water may vary depending upon the sources, way of collection and the treatment. Although the wastewater contains large proportions of organic matter and essential nutrients of plants, toxic metals present in it will be harmful to ecosystem, environment and public health [12]. Heavy metals such as cobalt (Co), cadmium (Cd), chromium (Cr), and arsenic (As) were deliberated as cancer-causing agents while nickel, iron, zinc, copper and manganese were deliberated as vital trace elements. However, the intake of heavy metals through polluted forage crops might pretense a danger to human health via meat consumption. The trace elements become harmful to our health if the levels of these essential elements are above the acceptable limits in the water, soil, and plants [2,4,9]. The toxicity of heavy metals in the body can cause renal, visual and immunological, musculoskeletal and generative effects [15].

## 5. Conclusions

We concluded that bioaccumulation of cobalt (Co) was significantly higher but below the permissible limits, and further irrigation with sewage water should be avoided. Highest concentration of Co was recorded in sheep blood, cow hair and sheep feces. Bioconcentration factor, pollution load index, enrichment factor and daily intake were higher in soil under cultivation of *S. bicolor*, *C. dactylon* and *S. fruticosa*. Highest concentration of cobalt was occurred in *S. bicolor* at site 1, in sheep blood, cow hair and sheep feces that were 0.545, 0.549 and 0.548 mg/kg, respectively, at site II and I. The general values were lower than the permissible maximum limits; it was observed that the bioaccumulation in the forage species was higher at some sites. High cobalt level in forages, particularly *S. bicolor* and *C. dactylon* from different sites, calls for the sound management of hazardous wastewater. High levels of Co toxicity showed the possible health risks to cattle and the human population (through animal meat) of the studied region. It is highly needed to address the cobalt contamination in the soil–plant–water–livestock nexus to avoid its entry into the food chain of the domesticated animals.

**Author Contributions:** Z.I.K. and K.A. conceived and designed the study. M.I.H., M.S.A., M.S.E. critically revised the manuscript. All authors approved the final version. M.N. executed the experiment and compiled data. M.U.F.A. statistically analyzed the data and help in chemical analysis. Z.I.K. critically edited and revised the manuscript, prepared the graphs. M.N. helped in sample collection and chemical analysis. All authors have read and agreed to the published version of the manuscript.

**Funding:** We acknowledge the Higher Education Commission of Pakistan for their financial cooperation in this research project #2484/13.

**Institutional Review Board Statement:** Departmental Ethical Review Committee provided ethical approval to conduct study.

**Informed Consent Statement:** Informed consent was taken from formers to conduct the study and to collect the samples. They were briefed about the research plan in details.

**Data Availability Statement:** Data and material is available for research purpose and for reference.

**Acknowledgments:** The authors extend their appreciation to the Researchers supporting project number (RSP-2021/173) King Saud University, Riyadh, Saudi Arabia.

**Conflicts of Interest:** The authors declare that they have no conflict of interest.

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
