# Peer review of "Blood, Hair and Feces as an Indicator of Environmental Exposure of Sheep, Cow and Buffalo to Cobalt: A Health Risk Perspectives"

_sustainability, doi:10.3390/su13147873_

Round 1
Reviewer 1 Report
Thank you for the opportunity to review this paper.
The study presented is complex. The authors present a major problem, that of Co contamination of plants and implicitly of animals. It follows the path of Co contamination from wastewater from soils, crops and animals. The risk to human health is also discussed, as the several human sickness are directly correlated with metal intoxication that enter the food chain through water-soil-plant / animal ecosystem.
I recommend this paper be accepted and published in this journal. However, there are some recommendations regarding this manuscript.
Point 1: Line 181 - deleted point befor citation and entered after citation
Point 2: Lines 190, 248 - entered point at the end of the sentence
Point 3: Line 212 - deleted point befor citation
Point 4: Line 209 - entered point after mm and befor 1
Point 5: Line 222 - deleted supplementary points
Point 6: Lines 225, 228 - insert space between words
Point 7: Lines 251, 368 and All the manuscript please check how the units of measure are written according to the templates
Point 8: Lines 267, 335, 336 - entered space between number and word
Point 9: Lines 388, 389, 390, 422, 424, 425, 445, 449 insert space where necessary
Point 10: In all the manuscript please cited uniformly site-I, site-II, site-III
Point 11: Reviewing and correcting in the whole document of the way of quoting the authors
Point 12: Reviewing and correcting the citation of references in accordance with the instructions in the templates

Author Response
Chief Editor
Sustainability MDPI
Dear editor
Reference to your email dated June 29th, 2021 regarding the Minor Revision of Manuscript ID: sustainability-1278121, entitled: Blood, hair and feces as an indicator of environmental exposure of sheep, cow and buffalo to cobalt: A health risk perspectives
Thank you for the most helpful review of our manuscript that has been submitted to the Sustainability MDPI.
We believe that your suggestions have greatly improved the manuscript and argued us to clarify any issue of ambiguity. We have responded to nearly all the suggestions.
Below please find our responses for all suggestions/comments (our responses are in red fonts). In addition, the changes that made on the text are also highlighted in RED font.
If you need any further clarification, please let us know.
Yours sincerely
- Iftikhar Hussain
Corresponding Author
Reviewer Report I
Comments and Suggestions for Authors
Thank you for the opportunity to review this paper.
The study presented is complex. The authors present a major problem, that of Co contamination of plants and implicitly of animals. It follows the path of Co contamination from wastewater from soils, crops and animals. The risk to human health is also discussed, as the several human sickness are directly correlated with metal intoxication that enter the food chain through water-soil-plant / animal ecosystem.
I recommend this paper be accepted and published in this journal. However, there are some recommendations regarding this manuscript.
Point 1: Line 181 - deleted point befor citation and entered after citation
- It has been corrected.
Point 2: Lines 190, 248 - entered point at the end of the sentence
- It has been corrected.
Point 3: Line 212 - deleted point befor citation
- It has been corrected.
Point 4: Line 209 - entered point after mm and befor 1
- It has been corrected.
Point 5: Line 222 - deleted supplementary points
- It has been corrected.
Point 6: Lines 225, 228 - insert space between words
- It has been corrected.
Point 7: Lines 251, 368 and All the manuscript please check how the units of measure are written according to the templates
- It has been corrected.
Point 8: Lines 267, 335, 336 - entered space between number and word
- It has been corrected.
Point 9: Lines 388, 389, 390, 422, 424, 425, 445, 449 insert space where necessary
- It has been corrected.
Point 10: In all the manuscript please cited uniformly site-I, site-II, site-III
- The sites are uniformly cited in the whole manuscript.
Point 11: Reviewing and correcting in the whole document of the way of quoting the authors
- The citations of authors have been changed according to the journal style.
Point 12: Reviewing and correcting the citation of references in accordance with the instructions in the templates
- The references have been arranged according to the journal style.

Reviewer 2 Report
The manuscript entitled "Blood, hair and feces as an indicator of environmental exposure of sheep, cow and buffalo to cobalt: A health risk perspectives" investigates the influence of sewage water irrigation on cobalt (Co) toxicity and bio accumulation in plant-soil environment and assess the health risk of grazing livestock via forage consumption. Although I really appreciated the title, which entices you to read the article, I believe that in its current state, the paper presents many aspects to be reviewed. The first refers to the introduction. I suggest to the authors to rewrite it because several sentences are unclear, others could be eliminated (lines 55-58) and the English needs to be improved. In this form it is sometimes boring. Secondly, same problems are present in materials and methods. The sentences in line 152-153,164-166, and 172-173 are not clear. In order to improve the quality of the paper it is fundamental to include information on the waste water treatments, its characterization and on the criteria used to define the 5 indicators studied. The term test is used in inappropriate way and must be replaced with sample. Several inaccuracies were found in the results: the abbreviation of cobalt is confused with that of organic carbon (CO), the sentence in line 288-290 does not refer to figure 2 but to 3 and the tables could be enriched with statistical analysis. The discussions paragraph has been written superficially and several references are no exactly appropriate (lines 384-386, 401-408 and 472-480). The sentence in line 480-485 was written twice in the manuscript. Authors could move the first part (lines 347-372) in the introduction.
As regards the bibliography, the formatting needs to be corrected. On this base the paper needs a strong revision.
In the end, some specific questions for the authors:
1) why was the soil first air evaporated and then extra dried in the stove for 72 hours? What do the authors mean by crusher soil test?
2)what do the authors mean with "all the natural issue" in line 216 and "particular scavenger" in line 184-185?
3) what statistical tests were used?
4) the abbreviation HMs is assumed to correspond to heavy metals. it's right? if so, it would be specified in the text.
Author Response
Reviewer Report III
Comments and Suggestions for Authors
The manuscript entitled "Blood, hair and feces as an indicator of environmental exposure of sheep, cow and buffalo to cobalt: A health risk perspectives" investigates the influence of sewage water irrigation on cobalt (Co) toxicity and bio accumulation in plant-soil environment and assess the health risk of grazing livestock via forage consumption.
Although I really appreciated the title, which entices you to read the article, I believe that in its current state, the paper presents many aspects to be reviewed. The first refers to the introduction. I suggest to the authors to rewrite it because several sentences are unclear, others could be eliminated (lines 55-58) and the English needs to be improved.
- I have deleted the lines 55 – 58 from the Introduction that is significantly revised and improved. Flow of information in the introduction section has been improved for clarity. The manuscript has been elaborated with recent literatures, focused on well-defined questions. In addition, the objectives of the work and novelty of the manuscript have been highlighted.
- The English language and grammar has been checked by a native speaker and corrected throughout the manuscript.
In this form it is sometimes boring. Secondly, same problems are present in materials and methods. The sentences in line 148 – 149, 152-153,164-166, and 172-173 are not clear.
- I have rephrased all these specified sentences and one whole paragraph (sample collection) in the material & methods.
- In order to improve the quality of the paper it is fundamental to include information on the waste water treatments, its characterization and on the criteria used to define the 5 indicators studied.
- The manuscript has been elaborated with recent literatures, focused on well-defined questions. In addition, the objectives of the work and novelty of the manuscript have been highlighted. The selected environmental and health risks indices were selected as reported in the literature and in our previous published manuscripts on other crops and vegetables.
- Please note that we do not have data for wastewater treatment details. Moreover, the wastewater treatment details were not the objectives of this study.
The term test is used in inappropriate way and must be replaced with sample.
- The term “test” has been replaced with “samples” in the whole manuscript.
Several inaccuracies were found in the results: the abbreviation of cobalt is confused with that of organic carbon (CO).
- It has been corrected as Co in the whole manuscript.
the sentence in line 288-290 does not refer to figure 2 but to 3 and the tables could be enriched with statistical analysis.
- We have changed and verified the results according to your suggestions. The statistical analysis and test details have been added in the Ms. All the pollution and health risk indices were calculated from the basic experimental data.
The discussions paragraph has been written superficially and several references are no exactly appropriate (lines 384-386, 401-408 and 472-480). The sentence in line 480-485 was written twice in the manuscript. Authors could move the first part (lines 347-372) in the introduction.
- We have modified, corrected and revised all these lines in the Ms according to your suggestions
- Repeated text (line 472 -480) has been deleted.
- Meanwhile, flow of information in the introduction section has been improved for clarity. The manuscript has been elaborated with recent literatures, focused on well-defined questions. In addition, the objectives of the work and novelty of the manuscript have been highlighted.
As regards the bibliography, the formatting needs to be corrected. On this base the paper needs a strong revision.
- All the references and bibliography in the Ms text and in the reference list have been revised, formatted and arranged according to the journal style.
In the end, some specific questions for the authors:
- why was the soil first air evaporated and then extra dried in the stove for 72 hours? What do the authors mean by crusher soil test?
- These typing mistakes have been removed. We have modified, corrected and revised all these lines in the M & M.
2)what do the authors mean with "all the natural issue" in line 216 and "particular scavenger" in line 184-185?
- These typing mistakes have been removed. We have modified, corrected and revised all these lines in the M & M.
- what statistical tests were used?
- I have added the missing information. We have used post hoc Tukey’s HSD test.
4) the abbreviation HMs is assumed to correspond to heavy metals. it's right? if so, it would be specified in the text.
Yes, HM is used for heavy metals in this manuscript. It has been specified in the text.

Reviewer 3 Report
The authors propose a manuscript titled “Blood, hair and feces as an indicator of environmental exposure of sheep, cow and buffalo to cobalt: A health risk perspectives”. The study take into consideration the exposure to toxic metals (TMs) such as cobalt that can cause lifelong carcinogenic disorder and mutagenic outcomes, and was evaluate the influence of sewage water irrigation on cobalt (Co) toxicity and bioaccumulation in plant-soil environment and to assess the health risk of grazing livestock via forage consumption. It’s important remembered that Cobalt (Co) is very necessary element for the growth of plants and animals, however, higher concentration have toxic impacts. The work is original with interesting data and worthy of applications in the agronomic field, but some further crucial notions, easy to add, are necessary for its publication.
- Introduction
Please choose a references for this statements and add a crucial concept in the suggested way:
- Lines 39-41. “Agriculture sector need significant amount of fresh water to increase agriculture production (vegetables, fruit, cereal, oil seed crops, legumes) in order to achieve a country’s food security and resilience (REFERENCE).
- Lines 55-58. Metals are substances with high electrical conductivity, adaptability, and gleam, which purposefully lose their electrons to shape cations (REFERENCE). Metals are found typically on the planet's outside and their courses of action move among different regions, achieving spatial assortments of enveloping core interests (REFERENCE).
- Lines 64-65. Heavy metals are a vital source of environment degradation and biodiversity loss (Afzal et al., 2020), to which plants, and in particular vegetation, seem to reply specifically (Perrino et al. 2014).
- Lines 143-144. When the authors reporting for the first time the scientific plant name, please consider their complete name including the author in the suggested way: e.g.
- Sorghum bicolor Kuntze
- Sesbania bispinosa (Jacq.) W. Wight
- Cynodon dactylon (L.) Pers.
- Pseudo fruticose No valid name, which is the correct name of this species?
- Tribulus terrestris instead Tribulus terresteris
Reference to be added:
- Perrino, E.V.; Brunetti, G.; Farrag, K. Plant communities of multi-metal contaminated soils: a case study in National Park of Alta Murgia (Apulia Region - southern Italy). Int J Phytoremediation, 2014, 16, 871-888. Doi: 10.1080/15226514.2013.798626.
- Materials and Methods
- Please specify the geographical system utilized (WGS84?)
- If possible give a better map of study area. Now is a not clear immage
- Line 169. See my previous comment on Pseudo fruticosa and Tribulus terresteris.
- Line 171. Cynodon dactylon instead Cynadaon dactylon
- Results
Well done. Few observations.
- Line 285-286. terrestris not terresteris.
- Lines 268, 286, 287…. Now I read fruticosa, so if is correct check whole document and change Pseudo fruticosa with P. fruticosa (=Phlomis fruticosa)
- Discussion
Well done. Few observations
- Lines 382-384. Complete in the suggested way. Metals, for example, Fe and Mn are primarily of lithological cause, however unreasonable metal substance, for example, Cr, Cd, Cu, Pb, Zn (Martin et al., 2013) is the responsibility of anthropogenic activities, although the Zn is a useful microelement for humans that occurs naturally in some ancestors of cultivated wheat(Perrino and Wagensommer 2021,Velu et al. 2018).
Reference to be added:
- Velu, G.; Singh, R.P.; Crespo-Herrera, L.; Juliana, P.; Dreisigacker, S.; Valluru, R.; Stangoulis, J. Singh Sohu, V.; Singh Mavi, G.; Mishra, V.K.; et al. Genetic dissection of grain zinc concentration in spring wheat for mainstreaming biofortification in CIMMYT wheat breeding. Rep. 2018, 8, 13526, doi:10.1038/s41598-018-31951-z.
- Perrino, E.V.; Wagensommer, R.P. Crop Wild Relatives (CWR) Priority in Italy: Distribution, Ecology, In Situ and Ex Situ Conservation and Expected Actions. Sustainability, 2021, 13, 4, 1682. https://doi.org/10.3390/su13041682
- Line 464. See my previous comment to scientific name. Phaseolus vulgaris….
- Conclusion
Please spend to more words for the results and future prospective
References
Please see the guidelines of the journal. Also there are many mistakes, e.g. 58 refer to which authors, also reporting doi in the correct way, others…
Author Response
Reviewer Report II
Comments and Suggestions for Authors
The authors propose a manuscript titled “Blood, hair and feces as an indicator of environmental exposure of sheep, cow and buffalo to cobalt: A health risk perspectives”. The study take into consideration the exposure to toxic metals (TMs) such as cobalt that can cause lifelong carcinogenic disorder and mutagenic outcomes, and was evaluate the influence of sewage water irrigation on cobalt (Co) toxicity and bioaccumulation in plant-soil environment and to assess the health risk of grazing livestock via forage consumption. It’s important remembered that Cobalt (Co) is very necessary element for the growth of plants and animals, however, higher concentration have toxic impacts. The work is original with interesting data and worthy of applications in the agronomic field, but some further crucial notions, easy to add, are necessary for its publication.
- Introduction
Please choose a references for this statements and add a crucial concept in the suggested way:
- Lines 39-41. “Agriculture sector need significant amount of fresh water to increase agriculture production (vegetables, fruit, cereal, oil seed crops, legumes) in order to achieve a country’s food security and resilience (REFERENCE).
- The references have been included and arranged according to the journal style.
- Lines 55-58. Metals are substances with high electrical conductivity, adaptability, and gleam, which purposefully lose their electrons to shape cations (REFERENCE). Metals are found typically on the planet's outside and their courses of action move among different regions, achieving spatial assortments of enveloping core interests (REFERENCE).
- The missing references have been included in these sentences according to the journal style.
- Lines 64-65. Heavy metals are a vital source of environment degradation and biodiversity loss (Afzal et al., 2020), to which plants, and in particular vegetation, seem to reply specifically (Perrino et al. 2014).
- The suggested sentence and references have been included in these sentences according to the journal style.
- Lines 143-144. When the authors reporting for the first time the scientific plant name, please consider their complete name including the author in the suggested way: e.g.
- Sorghum bicolor Kuntze
- Sesbania bispinosa (Jacq.) W. Wight
- Cynodon dactylon (L.) Pers.
- Pseudo fruticose No valid name, which is the correct name of this species?
- Tribulus terrestris instead Tribulus terresteris
- Thanks for your comments. The full scientific names of experimental plants have been corrected in the whole manuscript, in Tables and in the figures. Meanwhile, names of forages have also corrected throughout Ms.
- Reference to be added:
- Perrino, E.V.; Brunetti, G.; Farrag, K. Plant communities of multi-metal contaminated soils: a case study in National Park of Alta Murgia (Apulia Region - southern Italy). Int J Phytoremediation, 2014, 16, 871-888. Doi: 10.1080/15226514.2013.798626.
- The suggested reference has been included in these sentences according to the journal style.
- Materials and Methods
- Please specify the geographical system utilized (WGS84?)
- Geographically, Toba-Tek-Singh exhibit 30°33' to 31°2' N and the longitude 72°08' to 72°48' E. It is included in the Ms.
- If possible give a better map of study area. Now is a not clear image
- The new improved figure 1 has been included in the Ms.
- Line 169. See my previous comment on Pseudo fruticosa and Tribulus terresteris.
- Line 171. Cynodon dactylon instead Cynadaon dactylon
- These minor corrections have been done in the Ms in the specified places.
- Results
Well done. Few observations.
- Line 285-286. terrestris not terresteris.
- It has been corrected.
- Lines 268, 286, 287…. Now I read fruticosa, so if is correct check whole document and change Pseudo fruticosa with P. fruticosa (=Phlomis fruticosa)
- The tested forage is Suaeda fruticosa. The full scientific names of experimental plants have been corrected in the whole manuscript, in Tables and in the figures
- Discussion
Well done. Few observations
- Lines 382-384. Complete in the suggested way. Metals, for example, Fe and Mn are primarily of lithological cause, however unreasonable metal substance, for example, Cr, Cd, Cu, Pb, Zn (Martin et al., 2013) is the responsibility of anthropogenic activities, although the Zn is a useful microelement for humans that occurs naturally in some ancestors of cultivated wheat (Perrino and Wagensommer 2021,Velu et al. 2018).
- The suggested sentence and references have been included in these sentences according to the journal style.
Reference to be added:
- Velu, G.; Singh, R.P.; Crespo-Herrera, L.; Juliana, P.; Dreisigacker, S.; Valluru, R.; Stangoulis, J. Singh Sohu, V.; Singh Mavi, G.; Mishra, V.K.; et al. Genetic dissection of grain zinc concentration in spring wheat for mainstreaming biofortification in CIMMYT wheat breeding. Rep. 2018, 8, 13526, doi:10.1038/s41598-018-31951-z.
- Perrino, E.V.; Wagensommer, R.P. Crop Wild Relatives (CWR) Priority in Italy: Distribution, Ecology, In Situ and Ex Situ Conservation and Expected Actions. Sustainability, 2021, 13, 4, 1682. https://doi.org/10.3390/su13041682
- The suggested sentence and references have been included in these sentences according to the journal style.
- Line 464. See my previous comment to scientific name. Phaseolus vulgaris….
- It has been corrected as Phaseolus vulgaris L.
- Conclusion
Please spend to more words for the results and future prospective.
- The conclusion has been revised.
References
Please see the guidelines of the journal. Also there are many mistakes, e.g. 58 refer to which authors, also reporting doi in the correct way, others…
- The references have been arranged according to the journal style.

Round 2
Reviewer 2 Report
The manuscript was strongly revised by the authors. I repeat that the sentences in lines 247-248 and 264-266 do not refer to the tables shown in brackets. I suggest further improving the formatting of the bibliography. However, in this current state, the manuscript can be accepted.